# Acoustic Voice Analysis as a Useful Tool to Discriminate Different ALS Phenotypes

**DOI:** 10.3390/biomedicines11092439

**Published:** 2023-08-31

**Authors:** Giammarco Milella, Diletta Sciancalepore, Giada Cavallaro, Glauco Piccirilli, Alfredo Gabriele Nanni, Angela Fraddosio, Eustachio D’Errico, Damiano Paolicelli, Maria Luisa Fiorella, Isabella Laura Simone

**Affiliations:** 1Neurology Unit, Department of Translational Biomedicine and Neurosciences, 70121 Bari, Italy; giammarco.milella91@gmail.com (G.M.); glauco.piccirilli@gmail.com (G.P.); alfredo.gabriele.nanni@gmail.com (A.G.N.); anfraddo@libero.it (A.F.); eust1980@libero.it (E.D.); damiano.paolicelli@uniba.it (D.P.); 2Otolaryngology Unit, Department of Translational Biomedicine and Neurosciences (DiBraiN), University of Bari Aldo Moro, 70121 Bari, Italy; log.sciancalepore@gmail.com (D.S.); giadacavallaro@live.it (G.C.); marialuisa.fiorella@uniba.it (M.L.F.); 3School of Medicine, University of Bari Aldo Moro, 70124 Bari, Italy

**Keywords:** ALS, voice analysis, ALS phenotypes, bulbar impairment

## Abstract

Approximately 80–96% of people with amyotrophic lateral sclerosis (ALS) become unable to speak during the disease progression. Assessing upper and lower motor neuron impairment in bulbar regions of ALS patients remains challenging, particularly in distinguishing spastic and flaccid dysarthria. This study aimed to evaluate acoustic voice parameters as useful biomarkers to discriminate ALS clinical phenotypes. Triangular vowel space area (tVSA), alternating motion rates (AMRs), and sequential motion rates (SMRs) were analyzed in 36 ALS patients and 20 sex/age-matched healthy controls (HCs). tVSA, AMR, and SMR values significantly differed between ALS and HCs, and between ALS with prevalent upper (pUMN) and lower motor neuron (pLMN) impairment. tVSA showed higher accuracy in discriminating pUMN from pLMN patients. AMR and SMR were significantly lower in patients with bulbar onset than those with spinal onset, both with and without bulbar symptoms. Furthermore, these values were also lower in patients with spinal onset associated with bulbar symptoms than in those with spinal onset alone. Additionally, AMR and SMR values correlated with the degree of dysphagia. Acoustic voice analysis may be considered a useful prognostic tool to differentiate spastic and flaccid dysarthria and to assess the degree of bulbar involvement in ALS.

## 1. Background

Amyotrophic lateral sclerosis (ALS) is a fatal neurodegenerative disorder characterized by progressive degeneration of upper (UMNs) and lower motor neurons (LMNs) in the cortex, brainstem, and spinal cord [1]. Given the clinical heterogeneity and absence of pathognomonic findings, unraveling UMN and LMN impairment in all body districts is necessary to achieve higher diagnostic and prognostic accuracy. Several ALS diagnostic criteria have been previously proposed, ranging from exclusively clinical evaluations of UMN and LMN signs [2,3,4] to neurophysiological-based Awaji criteria [5].

However, the assessment of both UMNs and LMNs in the bulbar region remains challenging for expert neurologists due to the lack of accurate clinical features, as well as radiological and electrophysiological findings [6]. Indeed, while the evidence of pathological reflexes (e.g., brisk jaw jerk, gag, and other facial reflexes), and muscle weakness, atrophy, and fasciculations, suggests UMN and LMN impairment, respectively, one of the most common bulbar symptoms, namely dysarthria, is typically characterized by mixed spastic-flaccid paresis [7,8]. The spastic or pseudobulbar dysarthria, referred to as “harsh,” “strained,” or “strangled” voice quality, is typically linked to UMN dysfunction, associated with slow tongue movements, and brisk gag, facial, and jaw reflexes [9,10,11]. Conversely, the flaccid or bulbar type dysarthria, characterized by a “breathy” or weak voice, hypernasality, nasal emissions, and articulatory imprecision without changes in the speaking rate, is classically associated with LMN dysfunction, characterized by tongue, palatal, and facial weakness and wasting, and poor or absent gag, facial, and jaw reflexes [9,10,11]. The relative contribution of spasticity and flaccidity to the impairment of speech intelligibility varies across individuals. However, to date, no evident clinical and paraclinical measures have been developed to distinguish these latter voice patterns, and the assignment of “spastic” or “flaccid” dysarthria has usually been part of subjective clinical evaluation.

Several acoustic and articulatory kinematic measures have been used to examine patients with neuromuscular disorders such as ALS.

### 1.1. Vowel Space Area

The vowel space area (VSA) is a bi-dimensional space where the two formant frequencies of the vocal tract, namely F1 and F2, are represented [11]. F1 lays on the horizontal axis and is usually influenced by tongue body height; F2 lays on the vertical axis and is typically influenced by the anterior–posterior movement of the tongue. The triangular vowel space area (tVSA) is constructed using the Euclidean distance between F1 and F2 coordinates of the vowels /i/, /u/, and /a/.

tVSA is calculated in the following equation [12]:
[(F1 /i/ (F2 /a/ − F2 /u/) + F1 /a/ (F2 /u/ − F2 /i/) + F1 /u/ (F2 /i/ − F2 /a/)]/2).

Many studies have shown that the VSA is larger in more intelligible speech than in less intelligible speech [13,14,15]. This phenomenon corresponds to greater articulatory excursions and more distinct acoustic-articulatory vowel targets [16]. In anomalous conditions (e.g., dysarthria), the range of articulatory movements decreases, a phenomenon known as vowel formant centralization [17]. In this case, higher formant frequencies tend to decrease, while lower formant frequencies tend to increase [18]. VSA is expected to be compressed in dysarthria-related conditions compared to normal speech [19]. Formant centralization and VSA compression in patients with dysarthria have been demonstrated in several studies [12,20,21]. Relative to ALS disease, some studies reported a restricted VSA [22,23,24]. Specifically, a common finding in the literature is that the speech of individuals with ALS is characterized by smaller VSA relative to that of control speakers [23]. Turner and Tjaden [25], however, failed to find significant differences in VSA between neurologically healthy adults and ALS patients, although there was a trend for reduced VSA in ALS [26].

### 1.2. Alternating Motion Rate (AMR) and Sequential Motion Rate (SMR)

Diadochokinesis is the ability to perform rapidly repeating or alternating movements. The alternating motion rate (AMR) and sequential motion rate (SMR) are the two traditional tests of oral diadochokinesis used to assess motor speech production [27]. AMR involves a “single” syllable being repeated at the maximum rate, whereas SMR is a “sequence” of syllables repeated at the maximum rate. The syllables traditionally employed are /pa/, /ta/, and /ka/ for AMR, and a combined sequence of /pataka/ for SMR. Of these, /pa/ evaluates the function of the lips, /ta/ evaluates the function of the tongue tip, and /ka/ evaluates the function of the tongue dorsum [28]. Both tasks have been found to be sensitive to the measurement and diagnosis of speech disorders arising from stroke [29], progressive neurological disease [30], apraxia of speech [31], and traumatic brain injury [32]. They have been shown to correlate with perceptual ratings of severity and intelligibility in dysarthria [27,30,33]. Dysarthric ALS patients have shown a decreased diadochokinetic rate compared to those non-dysarthric at baseline [8].

Very recently, acoustic analysis of voice has been proposed as a useful tool to improve the early detection of ALS bulbar impairment before clinical evident signs and to determine indices of disease progression both in the early stages of diagnosis and during the disease course [34,35,36].

## 2. Aim of the Study

To date, no studies have tested acoustic metrics in relation to ALS clinical measures. The aim of the present study was to evaluate the role of the proposed vowel space area (VSA), alternating motion rate (AMR), and sequential motion rate (SMR) as reliable and useful acoustic measures able to discriminate different ALS phenotypes.

## 3. Materials and Methods

### 3.1. Subjects and Inclusion/Exclusion Criteria

Thirty-six ALS patients referred to the ALS tertiary center for motor neuron diseases in Apulia, Southern Italy, between 2021 and 2022, were consecutively recruited for the study. Written consent was obtained from all patients. Exclusion criteria included severely impaired systemic conditions, incapacity to give consent, severe dysarthria, anarthria, aphasic symptoms, cognitive dysfunctions, impaired syntactic comprehension, and hearing impairments. Demographic characteristics and clinical data were collected by experienced neurologists of the ALS team. Recorded variables included age at clinical evaluation, gender, site of onset (bulbar/spinal), and disease duration.

Twenty healthy controls (HCs) without a medical history of inflammatory, autoimmune, vascular, or neurodegenerative diseases and a family history of ALS were also recruited.

### 3.2. Clinical Evaluation

All patients were functionally evaluated using the ALS Functional Rating Scale—Revised (ALSFRS-r), which includes 12 items assessing bulbar (1st–3rd items), upper limbs (4th–6th items), lower limbs (7th–9th items), and respiratory symptoms (10th–12th items) [37]. The selective bulbar (ALSFRS-r-B) score was also calculated for each patient. Upper motor neuron (UMN) burden was evaluated using the Penn Upper Motor Neuron scale (PUMNS) [38], a 28-item scale ranging from 0 (normal) to a maximum of 32 for widespread/severe UMN involvement. It evaluates the bulbar region (scores 0–4), upper limbs (scores 0–7 bilateral), and lower limbs (scores 0–7 bilateral). Patients were considered as presenting prevalent UMN (pUMN) or prevalent LMN (pLMN) impairment using the median value as the cut-off, as performed elsewhere [39].

The Dysphagia Outcome and Severity Scale (DOSS), a 7-point scale, was used to systematically rate the functional severity of dysphagia [40] based on objective assessment through fibrolarynscopy. Further evaluation of the dysphagia was performed by administering the self-administered questionnaire Dysphagia Risk Assessment Scale (DRAS) [41], a 17-item scale, whose Italian version has already been validated [42].

### 3.3. Voice Analysis and Phono-Articulatory Evaluation

Voice recording was performed using a Samson Meteor Mic—USB Studio Condenser Microphone (frequency response of 20 Hz–20 kHz) placed 20 cm from the lips in a quiet room (<30 dB background noise) and connected to the Audacity 2.1.2 software. The recording was done in a mono channel with a sampling rate set at 44 kHz and 16 bits. The input volume was set to 70%.

### 3.4. Voice Analysis

The protocol proposed by Vizza et al. [12] was used in the present study. To evaluate the patients’ articulatory abilities, areas of the vowel triangles were analyzed. Each patient was instructed to perform a continuous and sustained vowel /a:/ to evaluate the Maximum Phonation Time. Patients able to pronounce a vowel for more than 5 s were asked to produce continuous and sustained vowels /a:/, /i:/, /u:/ for this duration. Subsequently, the triangular vowel space area was calculated using the F1 and F2 values of the sustained vowels /a:/, /i:/, /u:/ (tVSA). Areas were calculated using GeoGebra, v.6.0.775, an interactive geometry, algebra, statistics, and calculus application developed by Markus Hohenwarter et al. in 2001.

### 3.5. Phono-Articulatory Evaluation

Alternating motion rates (AMRs) and sequential motion rates (SMRs) were obtained after the aforementioned tasks. The AMR consisted of the rapid repetition of syllables /pa/, /ta/, /ka/ as fast and as long as patients could manage in one breath, without blurring them together or stumbling over them. To evaluate the SMR, the sequence /pataka/ was repeated by the patients, following the same instructions given for the AMR [27]. After explaining every task, the examiner gave the patient a demonstration. Lastly, the patient was asked to read phonetically balanced sentences containing the seven Italian vowels between the same unvoiced plosive consonants (/p/, /t/, /k/). In this way, each vowel was performed in the same phonetic context three times. The audio file was exported in WAV format. The vocal signal was analyzed using Praat (version 6.1.09), a computer software package for speech, phonetic, and voice analysis developed by Paul Boersma and David Weenink from the Institute of Phonetic Sciences, University of Amsterdam [43].

The AMR and SMR tasks were used to obtain the articulation rate (art rate in syllables/seconds) of each syllable and the sequence /pataka/. A selection of nine syllables was made, and the duration of the selection was recorded in seconds. The articulation rate was then calculated to one decimal place (nine syllables divided by duration) [27].

### 3.6. Statistical Analysis

Demographic and clinical variables are reported as the median (along with interquartile range) or frequencies (percentages) for continuous and categorical variables. Between-group comparisons in voice analysis parameters were performed using the Mann–Whitney U test for continuous variables. Bivariate models were computed using Spearman correlations to assess correlations between voice analysis parameters and demographic and clinical parameters.

ROC curves were derived to test the accuracy of voice analysis parameters in discriminating ALS patients with prevalent upper and lower motor neuron impairment.

The significance level for all tests was set at α ≤ 0.05. Given the exploratory aim of the study, no formal adjustment of alpha was performed.

## 4. Results

### 4.1. Study Population

The 36 ALS patients consisted of 24 males and 12 females, with a median age of 61 years (IQR 56–71). At the clinical evaluation, eight patients (22.2%) had a bulbar onset, seven (19.4%) had a spinal onset without clinical symptoms of bulbar impairment, and twenty-one (58.4%) had a spinal onset with bulbar symptoms developed during the disease course. None of them fulfilled the criteria for ALS-Frontotemporal dementia according to Strong criteria [44]. Median values of all ALSFRS-r scores, selective bulbar (ALSFRS-r-B) scores, PUMNS, and disease duration, all evaluated at sampling, are reported in Table 1.

The voice parameters, namely tVSA, AMR for each syllable (/pa/, /ta/, /ka/), and SMR for the sequence /pataka/, did not differ between male and female ALS patients, nor did they correlate with age or disease duration at clinical evaluation. Therefore, sex, age, and disease duration, all evaluated at clinical evaluation, were not included as covariates in between-group comparisons.

Healthy controls (HCs) were age- (median: 59, IQR: 57–61) and sex- (14 males and 6 females) matched to ALS patients.

### 4.2. Voice Parameter Differences between ALS Patients and HCs

Overall, ALS patients exhibited significantly lower values of tVSA compared to HCs (*p* = 0.0017) (Figure 1A).

ALS patients also showed significantly lower values of AMR for the syllables /pa/ (Figure 1B), /ta/ (Figure 1C), /ka/ (Figure 1D), and lower values of SMR for the sequence /pataka/ (Figure 1E), compared to HCs (*p* < 0.001 for all).

### 4.3. Voice Parameters and Prevalent UMN or LMN Impairment

ALS patients with pUMN impairment exhibited significantly lower values of tVSA compared to patients with pLMN impairment (*p* = 0.004) (Figure 2A). Furthermore, a negative correlation was found between PUMNS and tVSA (r_s_ = −0.504, *p* = 0.002).

Patients with pUMN impairment also exhibited significantly lower values of AMR for the syllables /pa/ (*p* = 0.04) (Figure 2B), /ta/ (*p* = 0.043) (Figure 2C), and /ka/ (*p* = 0.018) (Figure 2D), and lower values of SMR for the sequence /pataka/ (*p* = 0.022) (Figure 2E), compared to patients with pLMN impairment. Moreover, PUMNS negatively correlated with AMR values for the syllables /pa/ (r_s_ = −0.343, *p* = 0.044), /ta/ (r_s_ = −0.348, *p* = 0.045), and /ka/ (r_s_ = −0.436, *p* = 0.009), and with SMR values for the sequence /pataka/ (r_s_ = −0.411, *p* = 0.014).

Among all the voice analysis parameters, tVSA exhibited the highest accuracy in discriminating patients with pUMN from pLMN (AUC: 0.83, CI: 0.707–0.965, *p* < 0.001) (Figure 3A), compared to articulation rate for the syllables /pa/ (AUC: 0.709, CI: 0.525–0.89, *p* = 0.04) (Figure 3B), /ta/ (AUC: 0.68, CI: 0.5–0.871, *p* = 0.048) (Figure 3C), and /ka/ (0.724, CI: 0.55–0.91, *p* = 0.027) (Figure 3D), and the sequence /pataka/ (AUC: 0.719, CI: 0.54–0.90, *p* = 0.031) (Figure 3E).

### 4.4. Voice Parameters and Site of Onset

The tVSA did not differ between ALS patients presenting bulbar onset, spinal onset without bulbar symptoms, and spinal onset with bulbar symptoms (*p* > 0.05) (Figure 4A).

Conversely, ALS patients showed significant differences in diadochokinetic parameters according to the site of onset. Specifically, bulbar-onset ALS patients exhibited lower values of AMR for the syllables /pa/, /ta/, and /ka/, and the SMR values for the sequence /pataka/ compared to spinal-onset ALS patients with bulbar symptoms at clinical evaluation. Furthermore, spinal-onset ALS patients with bulbar symptoms exhibited significantly lower values of AMR for the syllables /pa/, /ta/, and /ka/, and lower values of SMR for the sequence /pataka/ compared to spinal-onset ALS patients without bulbar symptoms at clinical evaluation. Obviously, bulbar-onset ALS patients showed lower values of all the AMRs and SMR than those with spinal-onset ALS patients without bulbar symptoms at clinical evaluation (*p* < 0.001 for all the analyses) (Figure 4B–E).

### 4.5. Voice Parameters and Clinical Evaluations

No correlations were found between all the acoustic parameters and total ALSFRS-r score. Additionally, tVSA did not correlate with ALSFRS-r bulbar subscore, DOSS, and DRAS scales. Instead, both AMR and SMR showed a positive correlation with the ALSFRS-r bulbar subscore (articulation rate for the syllables /pa/ r_s_ = 0.554, *p* = 0.001; /ta/ r_s_ = 0.467, *p* = 0.005; /ka/ r_s_ = 0.479, *p* = 0.004; /pataka/ r_s_ = 0.427, *p* = 0.011) and DOSS scale (articulation rate for the syllables /pa/ r_s_ = 0.62, *p* < 0.001; /ta/ r_s_ = 0.57, *p* < 0.001; /ka/ r_s_ = 0.614, *p* < 0.001; /pataka/ r_s_ = 0.57, *p* < 0.001), and a negative correlation with the DRAS scale (articulation rate for the syllables /pa/ r_s_ = −0.57, *p* = 0.001; /ta/ r_s_ = −0.36, *p* = 0.038; /ka/ r_s_ = −0.434, *p* = 0.021; /pataka/ r_s_ = −0.57, *p* = 0.002).

## 5. Discussion

The impairment of motor functions that characterizes ALS disease includes language disorders, and approximately 80 to 96% of people with ALS become unable to speak during the disease progression [45]. Previous studies have already reported that voice analysis may be an important measure for detecting bulbar dysfunction considering that bulbar motor changes (e.g., impaired speech or swallowing) are the first symptoms in 30% of ALS patients, and they are present in all patients at a later stage [46,47].

The aim of our study was to assess the potential role of voice analysis in distinguishing ALS clinical phenotypes, specifically ALS patients with pUMN or pLMN involvement, as well as patients with bulbar onset, or spinal onset with or without bulbar signs. The findings demonstrated significant differences in various acoustic parameters between ALS patients and HCs, as well as distinct profiles between ALS patients with pUMN or pLMN impairment and according to the site of onset.

Comparing the voice parameters of ALS patients with HCs, ALS patients exhibited lower values of tVSA, indicating reduced spectral complexity in vocalization. This suggests that ALS disease leads to alterations in vocal characteristics, resulting in a narrower frequency spectrum. Additionally, ALS patients displayed slower articulation rates for analyzed syllables, indicating motor speech deficits in ALS patients compared to HCs. Our findings were in line with previous studies that demonstrated the role of voice analysis as an objective assessment tool to differentiate dysarthric patients from healthy individuals [17,24,32,48,49]. Regarding other studies that did not find differences among specific acoustic features between ALS patients and HCs, the possible explanation could lie in the phenotypes of enrolled ALS patients, which are not often reported in the clinical characteristics of the study population [23,25,50]. Indeed, in our study, we demonstrated that clinical phenotypes severely impact acoustic voice parameters.

Specifically, in terms of differentiating UMN and LMN involvement, the study revealed that patients with predominant UMN impairment exhibited both lower tVSA values and slower articulation rates for the analyzed syllables compared to patients with predominant LMN impairment. Our findings support earlier studies, which suggest that ALS patients with “spastic” dysarthria encounter significant slowness of movement, despite exhibiting little to no muscle wasting or weakness in their bulbar muscles. This slowness was more severe than those with pLMN [8]. These results contribute further evidence to support the influence of UMN involvement on speech motor control in ALS [8].

Moreover, the site of onset in ALS has been shown to influence the pattern and severity of motor impairment. In this study, we failed to detect significant differences in tVSA values among ALS patients with different sites of onset. This lack of differences can be attributed to the fact that tVSA is influenced by the decline in speech intelligibility rather than the speed of articulation [15,23]. In patients in the early-to-middle stages of ALS, as in our cohort, adaptive changes in tongue–jaw coupling were observed in response to the biomechanical and muscular alterations affecting the articulators, particularly the tongue. These adaptations in tongue–jaw coupling played a partial role in mitigating the negative impact of articulatory impairment on the clarity of vowel sounds, assessed by tVSA. However, as the disease progressed to the late stage, these adaptive changes became less apparent, leading to a significant overall reduction in vowel contrasts [51,52,53].

On the other hand, we found significant differences in the diadochokinetic measures between bulbar-onset and spinal-onset ALS patients. Specifically, patients with bulbar-onset ALS showed slower articulation rates in comparison to patients with spinal-onset ALS. Furthermore, within the spinal-onset group, patients who developed bulbar symptoms during the progression of the disease exhibited significantly lower values of AMR and SMR when compared to those who did not develop bulbar signs. These findings agree with previous studies that reported that the rate of speech declined faster in bulbar-onset patients than in spinal-onset patients [54], and that instrumental-based measuring of speech, such as that of the articulation rate, can detect early-onset bulbar symptoms of ALS [55,56,57,58].

Furthermore, our study revealed significant correlations between these measures and the ALSFRS-r bulbar subscore, as well as the DOSS and DRASS scales. These findings suggest that these acoustic measures have the potential to be used as valuable prognostic indicators for assessing bulbar deterioration in individuals with ALS. Additionally, it is worth noting that we did not find correlations between the alternating motion rate (AMR), sequential motion rate (SMR), and the ALSFRS-total score, or with the disease duration. This suggests that these specific acoustic parameters primarily capture changes related to bulbar function, rather than reflecting the overall progression and duration of the disease. Indeed, it is important to consider that the ALSFRS-total score encompasses a broader range of functional domains, such as limb weakness and respiratory function, which may not accurately reflect the speech motor control effect in individuals with bulbar-onset ALS [37,59].

These findings support the notion that monitoring the alternating motion rate (AMR) and sequential motion rate (SMR) can provide valuable insights into the progression of bulbar involvement and its impact on speech motor control in ALS patients. The assessment of these acoustic measures over time may help predict the rate and severity of speech deterioration in individuals with ALS.

Despite the valuable insights provided by this study, several limitations should be acknowledged. First, the sample size of the study was relatively small, which may limit the generalizability of the findings. A larger and more diverse sample would strengthen the robustness of the results and allow for subgroup analyses based on factors such as disease progression.

Second, the study was cross-sectional, lacking longitudinal data to track the progression of voice parameters and their relationship with clinical outcomes over time. Longitudinal studies would provide a more comprehensive understanding of how voice analysis measures change throughout the course of the disease and their predictive value for disease progression and patient outcomes.

Finally, in our study, we did not evaluate the cognitive dysfunction which, in ALS disease, could range from mild cognitive impairment to frontotemporal dementia, with a significative impact on language [60,61,62].

We can conclude that voice analysis shows promise as a non-invasive and objective method for characterizing motor speech deficits in ALS. The provision of quantitative measures of vocal characteristics has the potential to contribute to understanding of disease progression, to aid in clinical decision making, and to facilitate the development of targeted interventions for ALS patients. Continued research in this area can advance our understanding of the pathophysiology of ALS and improve the management and quality of life of individuals affected by this devastating disease.

## Figures and Tables

**Figure 1 biomedicines-11-02439-f001:**
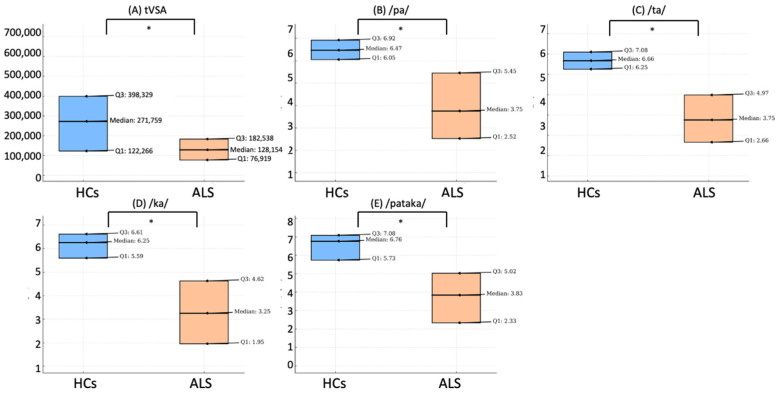
Comparison of voice parameters between healthy controls and ALS patients. This figure illustrates the differences in various voice parameters between healthy controls (HCs) and patients with amyotrophic lateral sclerosis (ALS). Box plot of tVSA (**A**); articulation rate of the syllable /pa/ (**B**); /ta/ (**C**); /ka/ (**D**); and the sequence /pataka/ (**E**). Each plot illustrates the median, first quartile (Q1), and third quartile (Q3) values, comparing the distributions between HCs and ALS patients. The symbol * denotes a significant difference at *p* < 0.05.

**Figure 2 biomedicines-11-02439-f002:**
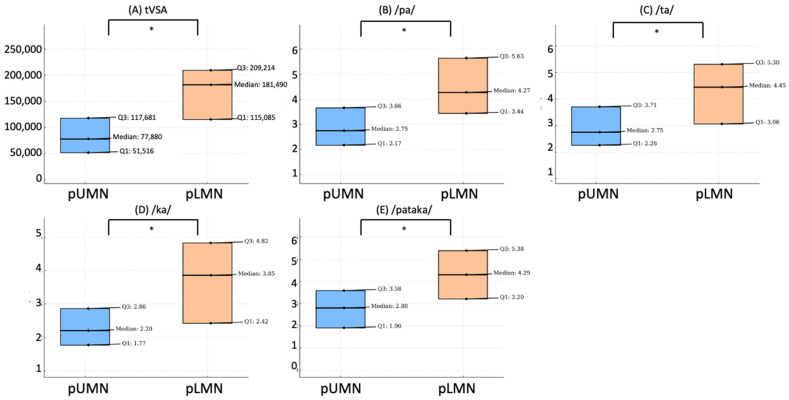
Comparison of voice parameters between prevalent UMN and LMN ALS patients. This figure shows the differences in voice parameters between ALS patients with prevalent upper motor neuron (pUMN) and prevalent lower motor neuron (pLMN). Box plot of tVSA (**A**); articulation rate of the syllable /pa/ (**B**); /ta/ (**C**); /ka/ (**D**); and the sequence /pataka/ (**E**). Each plot illustrates the median, first quartile (Q1), and third quartile (Q3) values, comparing the distributions between these two ALS phenotypes. The symbol * denotes a significant difference at *p* < 0.05.

**Figure 3 biomedicines-11-02439-f003:**
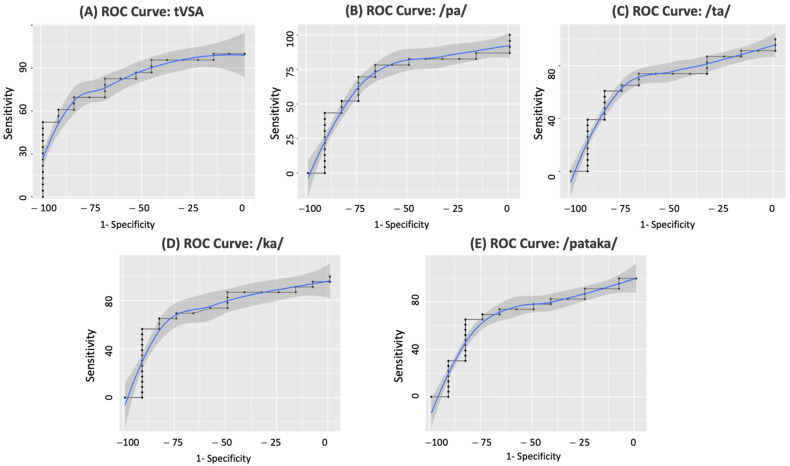
Discrimination of ALS patients with prevalent UMN and LMN impairment using voice parameters. The figure shows the ROC curves for various voice parameters able to discriminate ALS patients with prevalent upper motor neuron (pUMN) impairment and those with prevalent lower motor neuron (pLMN) impairment. ROC curve of tVSA (**A**); articulation rate of the syllable /pa/ (**B**); /ta/ (**C**); /ka/ (**D**); and the sequence /pataka/ (**E**). The area under each curve quantifies the accuracy of the respective voice parameter as a diagnostic tool for distinguishing these two ALS phenotypes.

**Figure 4 biomedicines-11-02439-f004:**
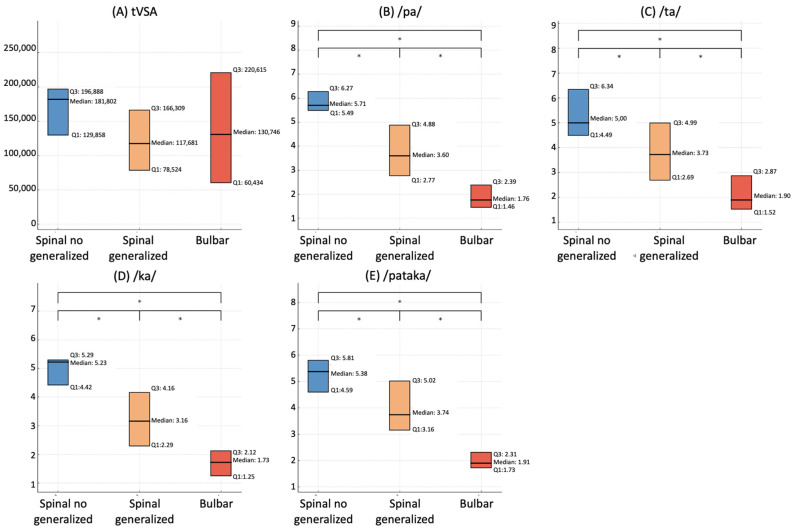
Comparison of voice parameters according to the site of onset in ALS patients. This figure illustrates the differences in various voice parameters among ALS patients, grouped according to their site of disease onset. Box plot of tVSA (**A**); articulation rate of the syllable /pa/ (**B**); /ta/ (**C**); /ka/ (**D**); and the sequence /pataka/ (**E**). Each plot illustrates the median, first quartile (Q1), and third quartile (Q3) values, comparing the distributions between patients with spinal onset (both with and without evidence of bulbar symptoms at clinical evaluation) and those with bulbar onset. The symbol * denotes a significant difference at *p* < 0.05.

**Table 1 biomedicines-11-02439-t001:** Demographic and clinical characteristics at sampling of ALS patients.

Total ALS Patients = 36		Median (IQR) or N. of Patients (%)
Age (years)		61 (56–71)
Sex	Female	12 (33.3%)
	Male	24 (66.7%)
Type of onset	Bulbar-onset	8 (22.2%)
	Spinal-onset with bulbar symptoms	21 (58.3%)
	Spinal-onset without bulbar symptoms	7 (19.4%)
Disease Duration (months)		30 (10–82)
Prevalent UMN/LMN impairment	Prevalent UMN	23 (63.9%)
	Prevalent LMN	13 (36.1%)
ALSFRS-R		34 (30–36)
ALSFRS-R bulbar sub score		9 (8–11)
Penn UMN Scale		13 (8–16)

Abbreviations: ALS: amyotrophic lateral sclerosis, UMN: upper motor neuron, LMN: lower motor neuron.

## Data Availability

The data that support the findings of this study are available from the corresponding author upon reasonable request.

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
