# Peer review of "Acoustic Voice Analysis as a Useful Tool to Discriminate Different ALS Phenotypes"

_biomedicines, 2023, doi:10.3390/biomedicines11092439_

Round 1

Reviewer 1 Report

biomedicines-2557867: “ACOUSTIC VOICE ANALYSIS AS A USEFUL TOOL TO DISCRIMINATE DIFFERENT ALS PHENOTYPES”

The authors attempt to demonstrate that some acoustic and articulatory kinematic measures might be useful in the testing of ALS patients looks pretty successful. An overcoming of the limitations mentioned by authors (relatively small sample size and a lack of longitudinal data) seemingly will allow full opening the potential of this approach. The material is described successively but results need to be reformatted.

Remarks/recommendations:

1)       Figures should be included into the text as closer as possible to their first references;

2)       in Figures, the plates should be denoted by letters;

3)       in Figures, significant differences should be denoted by asterisks or something suitable;

4)       the above changes allow the reformatting of the “Result” section both to exclude from the text a duplication of the data presented in Figures and to clarify the main results;

5)       in Figure 3, the fonts should be bigger;

6)       in line 23, “   HCs, and    “;

7)       in line 43, “…findings [6].”

Author Response

Reviewer 1

R: biomedicines-2557867: “ACOUSTIC VOICE ANALYSIS AS A USEFUL TOOL TO DISCRIMINATE DIFFERENT ALS PHENOTYPES”

The authors attempt to demonstrate that some acoustic and articulatory kinematic measures might be useful in the testing of ALS patients looks pretty successful.

A: We thank the reviewer for his appreciation.

R: An overcoming of the limitations mentioned by authors (relatively small sample size and a lack of longitudinal data) seemingly will allow full opening the potential of this approach. The material is described successively but results need to be reformatted.

A: In the present response, we have endeavored to overcome the limitations pointed out by the reviewer.

R: Remarks/recommendations: 

1)       Figures should be included into the text as closer as possible to their first references;

2)       in Figures, the plates should be denoted by letters;

3)       in Figures, significant differences should be denoted by asterisks or something suitable;

4)       the above changes allow the reformatting of the “Result” section both to exclude from the text a duplication of the data presented in Figures and to clarify the main results; 

5)       in Figure 3, the fonts should be bigger;

A: We sincerely thank the reviewer for his/her suggestions. Following the advice, in the “Result section”, we completely reformatted Figures 1-2-4, positioning them in the text as closer as possible to their first references, denoting them by letters, and marking the significant differences with asterisks. Accordingly, we modified the results in the text, removing duplicated information, such as the median and IQR values. Finally, we increased d the fonts in Figure 3.

R:        6)       in line 23, “   HCs, and    “;

7)       in line 43, “…findings [6].”

A: We corrected them according to the Reviewer’s suggestions.

Reviewer 2 Report

1. Content of manuscript - investigations of voice abnormalities in patients with ALS

2. Manuscript strenghts - parameters of physiological investigations: tVS, Alternating  and Sequential voice parameters

Weeknesses: small number of ALS patients, not known duration of ALS in a moment of investigation, not known cognitive deficit

3. Point-by-point list of recommendation for improvement:
    a) There is missing proper description of Figures
    b) There is not discussed duration of ALS, what can be important for the acustic findings
    c)Not correlation with cognitive deficits - what can influence speaking substrantially.
    d) The authors could not find differences between various site of origin ALS - these facts are to be more discussed.
    e) Valuable prognostic indicators - for assessing bulbar deterioration. And in ALS patients without bulbar symptoms - it can  add something to prognosis?

4. Minor facts for improvement.
Interpretation of findings in bulbar form and in spinal form - both have abnormal voice findings.
In results - too many facts with numbers and abbreviations - this is nearly not readible
Description of figures with advice of interpretation.

I understood perfectly, I have no comments.

Author Response

Reviewer 2

Comments and Suggestions for Authors

  1. Content of manuscript - investigations of voice abnormalities in patients with ALS
    2. Manuscript strenghts - parameters of physiological investigations: tVS, Alternating, and Sequential voice parameters

    Weeknesses: small number of ALS patients, not known duration of ALS in a moment of investigation, not known cognitive deficit

    3. Point-by-point list of recommendation for improvement:

R: a) There is missing proper description of Figures

A: a) We thank the reviewer for the suggestion. We added an extensive description under each single figure, as reported below.

“Figure 1. Comparison of Voice Parameters between Healthy Controls and ALS Patients.

This figure illustrates the differences in various voice parameters between healthy controls (HCs) and patients with Amyotrophic Lateral Sclerosis (ALS). The box plots depict the median, first quartile (Q1), and third quartile (Q3) of each parameter, comparing the distributions between HCs and ALS patients.”

“Figure 2. Comparison of Voice Parameters between UMN and LMN Dominant ALS Patients.

This figure shows the differences in voice parameters between ALS patients with prevalent Upper Motor Neuron (UMN) and prevalent Lower Motor Neuron (pLMN). The box plots illustrate the median, first quartile (Q1), and third quartile (Q3) of each parameter, highlighting the differences between these two ALS phenotypes.”

“Figure 3. Discrimination of ALS patients with prevalent UMN and LMN impairment using voice parameters.

The figure shows the ROC curves for various voice parameters able to discriminate ALS patients with prevalent Upper Motor Neuron (UMN) impairment and those with prevalent Lower Motor Neuron (LMN) impairment. The area under each curve quantifies the accuracy of the respective voice parameter as a diagnostic tool for distinguishing these two ALS phenotypes.”

“Figure 4. Comparison of Voice Parameters according to the site of onset in ALS patients

This figure illustrates the differences in various voice parameters among ALS patients, grouped according to their site of disease onset. The box plots depict the median, first quartile (Q1), and third quartile (Q3) of each parameter, comparing the distributions between patients with spinal onset (both with and without evidence of bulbar symptoms at clinical evaluation) and those with bulbar onset.”

R: b) There is not discussed duration of ALS, what can be important for the acustic findings

A: b) We thank the reviewer for the thoughtful comment.

We would like to point out that the disease duration was already reported in Table 1 (“Demographic and clinical characteristics of ALS patients”). In the “Result” section, we added a correlation between voice parameters and disease duration, and we did not find any correlation. In the “Discussion” section we remarked that Voice Parameters are influenced selectively by the degree of bulbar impairment, rather than by the overall progression and duration of the disease.

Below, the text was added in this regard in the “Result” and “Discussion” sections, respectively.

RESULTS-Study population” section: “The voice parameters, namely tVSA, AMR for each syllable (/pa/, /ta/, /ka/), and SMR for the sequence /pataka/, did not differ between male and female ALS patients, neither they correlate with age and disease duration at clinical evaluation. Therefore, age at clinical evaluation, sex and disease duration were not included as covariates in between-group comparisons.”

“DISCUSSION” section: “Additionally, it is worth noting that we have not found correlations between the Alternating Motion Rate (AMR), Sequential Motion Rate (SMR), and the ALSFRS-total score, as well as the disease duration. This suggests that these specific acoustic parameters primarily capture changes related to bulbar function, rather than reflecting the overall progression and duration of the disease”.

R:    c) Not correlation with cognitive deficits - what can influence speaking substrantially.

A:    c) We agree with the reviewer’s concerns regarding the impact of cognitive deficits on speaking abilities. Indeed, in the “Materials and methods- Subjects and Inclusion/Exclusion Criteria” we stated that “Exclusion criteria included severely impaired systemic conditions, incapacity to give consent, severe dysarthria, anarthria, aphasic symptoms, cognitive dysfunctions, impaired syntactic comprehension, and hearing impairments.”

In the “RESULTS- Study population” section, we further stressed this concept remarking that “None of ALS patients fulfilled the criteria for ALS-Frontotemporal dementia according to Strong criteria [44]. Finally, in the “Discussion” section we have already included the absence of an extensive cognitive evaluation of a limit of the study (“Finally, in our study, we did not evaluate the cognitive dysfunction which in ALS disease could range from mild cognitive impairment to frontotemporal dementia, with a significative impact on the language [60–62].”).

 R:   d) The authors could not find differences between various site of origin ALS - these facts are to be more discussed.

A: We thank the reviewer for the suggestions. We would like to remark that among all voice parameters analyzed in the work, only tVSA did not differ between patients with different site of onset. Based on the literature, we tried to explain this latter finding (“Discussion” section “This lack of differences can be attributed to the fact that tVSA is influenced by the decline in speech intelligibility rather than the speed of articulation [15, 23]. In patients in the early-to-middle stages of ALS, as in our cohort, adaptive changes in tongue-jaw coupling were observed in response to the biomechanical and muscular alterations affecting the articulators, particularly the tongue. These adaptations in tongue-jaw coupling played a partial role in mitigating the negative impact of articulatory impairment on the clarity of vowel sounds, assessed by tVSA. However, as the disease progressed to the late stage, these adaptive changes became less apparent, leading to a significant overall reduction in vowel contrasts [51–53].”.

On the contrary, we found that all the other voice parameters analyzed, namely AMR and SMR, showed significant differences among ALS patients with different sites of onset, namely bulbar and spinal, and among these latters those with bulbar symptoms exhibited significantly lower values than those without bulbar symptoms. In the “Discussion” section we tried to comment on this latter finding (“On the other hand, we found significant differences in the diadochokinetic measures between bulbar-onset and spinal-onset ALS patients. Specifically, patients with bulbar-onset ALS showed slower articulation rates in comparison to patients with spinal-onset ALS. Furthermore, within the spinal-onset group, patients who developed bulbar symptoms during the progression of the disease exhibited significantly lower values of AMR and SMR when compared to those who did not develop bulbar signs. These findings agree with previous studies which reported that the rate of speech decline faster in bulbar-onset patients than in the spinal-onset [54] and instrumental-based measuring of speech, such as that of articulation rate, can detect early-onset bulbar symptoms of ALS [55–58].”).

R:    e) Valuable prognostic indicators - for assessing bulbar deterioration. And in ALS patients without bulbar symptoms - it can  add something to prognosis?

A: We thank the reviewer for the appreciation and the suggestion. As regard the ALS patients with spinal-onset without bulbar symptoms at clinical evaluation, due to the small number of cases(n. 7),  we were not able to perform further subgroup analysis.

Actually, recruiting ALS patients with spinal-onset without bulbar symptoms could be extremely difficult, since when patients refer to a tertiary center, like ours, the median interval from symptom onset is already  12 months (DOI: 10.1016/j.jns.2006.06.027), and during this time patients with spinal-onset frequently develop bulbar symptoms. As stated in the limits of the study in the “Discussion” section, a larger and more diverse sample would allow for subgroup analyses, investigating the ability of voice parameters to predict the evidence of bulbar symptoms even in those patients without clinically evident involvement of bulbar districts.

R: 4. Minor facts for improvement. 
Interpretation of findings in bulbar form and in spinal form - both have abnormal voice findings.
In results - too many facts with numbers and abbreviations - this is nearly not readible
Description of figures with advice of interpretation.

A: We again thank the reviewer for the suggestions.

As specified above, as regard the site of onset (in the “Result” and “Discussion” sections), patients with bulbar-onset significantly differed from patients with spinal onset in the diadochokinetic measures, whereas no differences were found in the tVSA measure, and we tried to interpret these anomalies. In addition, we have already reported (in the “Result” and “Discussion” sections) that patients with spastic dysarthria, related to prevalent UMN impairment, differed from patients with flaccid dysarthria, related to LMN impartiment, differed in tVSA and diadochokinetic measures and we tried to interpret these anomalies.   

As stated above, we completely modify the figures and the descriptions of the figures. Furthermore, we modified the test in the “Results” section, in order to remove duplicated information (e.g., median values and IQR ranges).

Round 2

Reviewer 1 Report

biomedicines-2557867: “ACOUSTIC VOICE ANALYSIS AS A USEFUL TOOL TO DISCRIMINATE DIFFERENT ALS PHENOTYPES”

The authors have made a careful revision and responded to almost all points I raised.

Residual remarks:

1)     in Figures, each plate should be denoted by letter and described in the legend in Figures, each plate should be denoted by letter and described in the legend (this looks like formal requirement, however it allows the manipulations with the material easily in the text);

2)     in all Figures, the fonts on vertical axes and on the plates should be bigger;

3)     in the legends to all Figures, “ * denotes significant difference at p < 0.05” should be added;

4)     all previous Table and Figures should be removed from the text.

Author Response

The authors have made a careful revision and responded to almost all points I raised.

Residual remarks:

1)     in Figures, each plate should be denoted by letter and described in the legend in Figures, each plate should be denoted by letter and described in the legend (this looks like formal requirement, however it allows the manipulations with the material easily in the text);

2)     in all Figures, the fonts on vertical axes and on the plates should be bigger;

3)     in the legends to all Figures, “ * denotes significant difference at p < 0.05” should be added;

4)     all previous Table and Figures should be removed from the text.

We would like to express our gratitude to the Reviewer for their continued feedback. In response to the remaining remarks:

1)We have increased the size of the letters denoting each plate and provided a detailed description in the legend. Additionally, we have revised the text to clearly indicate the specific plate being referred to.

2)We have maximized the font size for both the vertical and horizontal axes.

3)We have incorporated the suggested sentence into the figures.

4)We have submitted a "clean version" of the manuscript, excluding the previous tables and figures.
